# BA.1/BA.5 Immunogenicity, Reactogenicity, and Disease Activity after COVID-19 Vaccination in Patients with ANCA-Associated Vasculitis: A Prospective Observational Cohort Study

**DOI:** 10.3390/v15081778

**Published:** 2023-08-21

**Authors:** Claudius Speer, Maximilian Töllner, Louise Benning, Marie Bartenschlager, Heeyoung Kim, Christian Nusshag, Florian Kälble, Marvin Reineke, Paula Reichel, Paul Schnitzler, Martin Zeier, Christian Morath, Wilhelm Schmitt, Raoul Bergner, Ralf Bartenschlager, Hanns-Martin Lorenz, Matthias Schaier

**Affiliations:** 1Department of Nephrology, University Hospital Heidelberg, 69120 Heidelberg, Germany; maximilian.toellner@googlemail.com (M.T.); louise.benning@med.uni-heidelberg.de (L.B.); christian.nusshag@med.uni-heidelberg.de (C.N.); florian.kaelble@med.uni-heidelberg.de (F.K.); marvin.reineke@gmail.com (M.R.); paulareichel@web.de (P.R.); martin.zeier@med.uni-heidelberg.de (M.Z.); christian.morath@med.uni-heidelberg.de (C.M.); matthias.schaier@med.uni-heidelberg.de (M.S.); 2Molecular Medicine Partnership Unit Heidelberg, EMBL, 69120 Heidelberg, Germany; 3Department of Infectious Diseases, Molecular Virology, Center for Integrative Infectious Disease Research, Medical Faculty Heidelberg, Heidelberg University, 68167 Heidelberg, Germany; marie.bartenschlager@med.uni-heidelberg.de (M.B.); heeyoung.kim@med.uni-heidelberg.de (H.K.); ralf.bartenschlager@med.uni-heidelberg.de (R.B.); 4Department of Infectious Diseases, Virology, University Hospital Heidelberg, 69120 Heidelberg, Germany; paul.schnitzler@med.uni-heidelberg.de; 5Center for Renal Diseases, 69469 Weinheim, Germany; w.schmitt@nierenzentrum-weinheim.de; 6Department of Internal Medicine A, Clinical Center Ludwigshafen, 67071 Ludwigshafen, Germany; bergnerr@klilu.de; 7German Center for Infection Research (DZIF), Heidelberg Partner Site, 69120 Heidelberg, Germany; 8Division Virus-Associated Carcinogenesis, German Cancer Research Center (DKFZ), 69120 Heidelberg, Germany; 9Division of Rheumatology, Department of Medicine V, University of Heidelberg, 69120 Heidelberg, Germany; hanns-martin.lorenz@med.uni-heidelberg.de

**Keywords:** ANCA-associated vasculitis, COVID-19, vaccination, omicron subtypes, neutralizing antibodies

## Abstract

Emerging omicron subtypes with immune escape lead to inadequate vaccine response with breakthrough infections in immunocompromised individuals such as Anti-neutrophil Cytoplasmic Antibody (ANCA)-associated vasculitis (AAV) patients. As AAV is considered an orphan disease, there are still limited data on SARS-CoV-2 vaccination and prospective studies that have focused exclusively on AAV patients are lacking. In addition, there are safety concerns regarding the use of highly immunogenic mRNA vaccines in autoimmune diseases, and further studies investigating reactogenicity are urgently needed. In this prospective observational cohort study, we performed a detailed characterization of neutralizing antibody responses against omicron subtypes and provided a longitudinal assessment of vaccine reactogenicity and AAV disease activity. Different vaccine doses were generally well tolerated and no AAV relapses occurred during follow-up. AAV patients had significantly lower anti-S1 IgG and surrogate-neutralizing antibodies after first, second, and third vaccine doses as compared to healthy controls, respectively. Live-virus neutralization assays against omicron subtypes BA.1 and BA.5 revealed that previous SARS-CoV-2 vaccines result in an inadequate neutralizing immune response in immunocompromised AAV patients. These data demonstrate that new vaccination strategies including adapted mRNA vaccines against epitopes of emerging variants are needed to help protect highly vulnerable individuals such as AAV patients.

## 1. Introduction

The Coronavirus Disease 2019 (COVID-19) pandemic caught the world unprepared and took a toll that many would have thought unimaginable before its occurrence in modern times. By early 2023, more than 750 million confirmed cases of COVID-19 have been reported to the World Health Organization (WHO) worldwide, including >6 million deaths [1,2]. Over the past three years, the COVID-19 pandemic has been gradually transitioned to an endemic state through exceptionally rapid vaccine development and attenuation of viral variants [3]. Real-world data afforded the protection by available vaccines and various vaccination regimens and show a clear correlation between the level of neutralizing antibodies after infection or vaccination and the frequency and severity of breakthrough infections [4,5]. However, there are still breakthrough infections with fatal outcomes in vulnerable populations such as elderly people or patients with severe pre-existing conditions and a compromised immune system due to an inadequate response to available vaccines [6,7,8]. In addition, while emerging variants of concern (VoCs) such as different omicron subtypes have resulted in significantly milder disease courses in the general population, their marked immune escape results in inadequate vaccine response with increased breakthrough infections in immunocompromised individuals [9,10,11,12,13,14].

A particularly vulnerable population comprises individuals with systemic autoimmune diseases such as anti-neutrophil cytoplasmatic antibody (ANCA)-associated vasculitis (AAV). To treat frequently life-threatening courses of this small-vessel vasculitis, a pronounced immunosuppressive induction and maintenance therapy is required, often including the use of the anti-CD20 monoclonal antibody rituximab, which is an established risk factor for severe COVID-19 courses [15]. Because of the immunosuppressive therapy and a frequently advanced age at initial AAV diagnosis, humoral and cellular immune responses after SARS-CoV-2 vaccination seem to be reduced and emerging omicron variants are of particular concern because of their high transmissibility and partial immune escape [16,17,18,19]. However, as AAV vasculitis is considered an orphan disease, there are still limited data on SARS-CoV-2 vaccination and prospective vaccination studies that have focused exclusively on AAV patients are lacking. In addition, there have been safety concerns regarding the use of new highly immunogenic mRNA vaccines in autoimmune diseases such as AAV vasculitis with regard to disease flares, and therefore, further studies on reactogenicity are urgently needed [20].

In this prospective observational cohort study, we provide an in-depth characterization of the humoral immune response after SARS-CoV-2 vaccination including live-virus assays to detect protective neutralizing antibodies against different omicron subtypes in AAV patients. We also longitudinally evaluate both vaccine reactogenicity and AAV disease activity to provide additional insight into the safety profile of previous SARS-CoV-2 vaccines.

## 2. Materials and Methods

### 2.1. Study Design

In this prospective, observational cohort study at 4 different German vasculitis centers (Division of Nephrology of the University Hospital of Heidelberg, Center for Renal Diseases Weinheim, Department of Internal Medicine A of the Clinical Center Ludwigshafen, Division of Rheumatology of the University Hospital of Heidelberg), we screened 140 AAV patients and healthy controls for participation in the study prior to SARS-CoV-2 vaccination. We included AAV patients with granulomatosis with polyangiitis (GPA), microscopic polyangiitis (MPA), and eosinophilic granulomatosis with polyangiitis (EGPA). Exclusively AAV patients with absence of disease activity were enrolled. We aimed to assess in-detail humoral immune response, live-virus neutralization against different SARS-CoV-2 variants, as well as reactogenicity and ANCA-associated vasculitis disease activity after a first, second, and third vaccine dose with mRNA and adenovirus-vectored SARS-CoV-2 vaccines. Individuals with reported COVID-19 were also excluded. In addition, before enrollment and after first, second, and third vaccine dose, antibodies to the nucleocapsid protein were measured, respectively. Individuals with positivity were excluded because of suspected recent SARS-CoV-2 infection and all individuals remained negative during follow-up (Figure 1). The healthy control cohort was matched for age and vaccine-type and ineligible subjects were excluded (Figure 1). A total of 64 AAV patients and 24 age- and vaccine-matched healthy controls were enrolled. After the initial standard two-dose SARS-CoV-2 vaccination, sera were available for 47 AAV patients and 16 healthy controls after the first vaccine dose and for all subjects after the second vaccine dose for both cohorts. A third booster vaccination with the mRNA vaccine BNT162b2 was performed in 21 AAV patients and all 24 healthy controls after a median (interquartile range, IQR) of 106 (79–124) days after second vaccination (Figure 1).

We determined anti-spike S1 IgG antibodies, SARS-CoV-2-specific surrogate-neutralizing antibodies, and antibodies to various SARS-CoV-2 epitopes three weeks after first and second vaccine dose in both cohorts, respectively. In addition, anti-spike S1 IgG antibodies, SARS-CoV-2-specific surrogate-neutralizing antibodies, and live-virus neutralization against recent SARS-CoV-2 variants was performed directly before and three weeks after a third vaccine dose (booster vaccination) in AAV patients and healthy controls, respectively. Adverse events and AAV disease activity were assessed three weeks after first, second, and third vaccine dose in AAV patients and healthy controls as described in detail below.

The study was approved by the Ethics Committee of the University of Heidelberg: S-416/2021 and conducted in accordance with the Declaration of Helsinki. Participants provided informed consent to participate in the study before taking part.

In 4 centers in Southwest Germany, 140 individuals were screened for participation in the study. Five ANCA-associated vasculitis patients were excluded due to prior COVID-19 infection as detected by positive nucleocapsid antibodies or positive SARS-CoV-2 PCR. A healthy control cohort was matched for age and vaccine, and subjects not eligible were excluded. A total of 64 ANCA-associated vasculitis patients were enrolled in the study, with available sera in 47, 64, and 21 patients after the first, second, and third vaccine dose, respectively. In addition, 24 age- and vaccine-matched healthy controls were included with available sera in 16, 24, and 24 subjects after the first, second, and third vaccine dose, respectively. * There was no recommendation for the timing of booster vaccination, especially not in immunocompromised individuals. Accordingly, the timing of booster vaccination in our study was inconsistent, and patients received their third vaccine dose partly at rheumatologists in private practice or at their general practitioner and were lost to follow-up.

### 2.2. Anti-SARS-CoV-2 S1 and Nucleocapsid IgG Chemiluminescent Immunoassay

Anti-spike S1 IgG antibodies were determined using the SARS-CoV-2 Total Assay (Siemens, Eschborn, German). Positivity was defined as a semi-quantitative index of ≥1, giving a specificity of 100% and a sensitivity of 100% for the detection of anti-S1 IgG antibodies. The Elecsys anti-SARS-CoV-2 assay (Roche, Mannheim, Germany) was performed to detect antibodies against the nucleocapsid protein and individuals with a positive result were excluded because of suspected recent SARS-CoV-2 infection. Both assays were performed according to the manufacturer’s instructions.

### 2.3. Detection of SARS-CoV-2-Specific Surrogate-Neutralizing Antibodies

A surrogate virus neutralization assay (Medac, Wedel, Germany) was performed to detect surrogate-neutralizing antibodies (nAB). Samples were pre-incubated with horseradish peroxidase (HRP) conjugated recombinant SARS-CoV-2 RBD fragment (HRP-RBD) to allow binding of the circulating neutralizing antibodies to the HRP-RBD. The pre-incubated samples were added to a capture plate, which was pre-coated with human ACE2 receptor protein (hACE2). Unbound HRP-RBD and any HRP-RBD with non-specific binding was captured on the plate, while HRP-RBD bound to neutralizing antibodies remained in the supernatant and was removed during washing. Optical density at 450 nm was measured in each well and the percentage (%) inhibition was calculated as follows:$$\mathrm{Inhibition}=\left(1-\left(\frac{\mathrm{OD}\;\mathrm{value}\;\mathrm{of}\;\mathrm{Sample}}{\mathrm{OD}\;\mathrm{value}\;\mathrm{of}\;\mathrm{Negative}\;\mathrm{Control}}\right)\right)\times 100\%\;$$

With a cut-off of ≥30% inhibition of RBD:ACE-2 binding, the assay achieves 99.9% specificity with 95–100% sensitivity for the detection of surrogate-neutralizing antibodies [21].

### 2.4. Bead-Based Multiplex Assay for the Detection of Antibodies to Various SARS-CoV-2 Epitopes

A multiplex bead-based assay for the Luminex platform (LabScreen COVID Plus) was performed (One Lambda Inc., West Hill, CA, USA) for the detection of IgG antibodies against various SARS-CoV-2 epitopes [22]. The assay includes the SARS-CoV-2 nucleocapsid protein, the SARS-CoV-2 full-spike protein, and the receptor-binding domain (RBD) of the spike protein. Antibody detection on antigen-coated microparticles was performed according to the manufacturer’s instructions and the mean fluorescence intensity (MFI) was analyzed on a Luminex 200 device (Luminex Corporation, Noord-Brabant, The Netherlands). The MFI cutoff values for each protein are listed in Appendix A.

### 2.5. Live-Virus Neutralization against the Omicron Subtypes BA.1 and BA.5

Neutralizing SARS-CoV-2 antibodies were determined in titration experiments on Calu3 cells as described previously by us [11]. SARS-CoV-2 virus stocks were produced by amplification of the BA.1 and BA.5 strain isolated from nasopharyngeal and oropharyngeal swabs of PCR-confirmed SARS-CoV-2-positive patients in Calu-3 cells. Two-fold serial dilutions of vaccine sera were incubated with the omicron subtypes BA.1 and BA.5. After 1 h at 37 °C, the mixture was added to VeroE6 cells and cells were fixed in the plates with 5% formaldehyde at 24 h post infection. Virus replication was determined by immunostaining for the viral nucleocapsid protein using an in-cell ELISA. Values were normalized to those obtained with cells infected in the absence of patient serum (100% infection) and non-infected cells (0% infection), the latter determining the assay background. The ID_50_ equates the serum dilution that reduces infection of cells by 50%. The cut-off for detection of this neutralization assay is at a neutralization titer of 1:10.

### 2.6. Monitoring of Reactogenicity and Disease Activity

Adverse events were assessed after first, second, and third vaccine dose in all AAV patients and healthy controls using a twelve-item questionnaire inquiring about previously mentioned side-effects after vaccination and the use of pain medication after vaccine reception. The questionnaire is given in the Appendix A. Local events included pain at the injection site, redness, and swelling. Systemic events included fever >38 °C, fatigue, headache, chills, muscle pain, joint pain, swollen lymph nodes, and others. The AAV disease activity was assessed via auto-antibodies against the ANCA target antigens myeloperoxidase (MPO) and proteinase 3 (PR3) and the kidney function, determined by serum creatinine and estimated glomerular filtration rate (eGFR). Both were assessed before vaccination as baseline and three weeks after first, second, and third vaccination, respectively.

### 2.7. Statistics

Data are expressed as median and IQR or number (N) and percent (%). Both groups were compared using the Mann–Whitney *U* test in case of continuous variables and the Fisher’s exact test in case of categorial variables. Results of more than two different groups were compared by applying the Kruskal–Wallis test with Dunn’s post-test. Statistical significance was assumed at a *p*-value < 0.05. The statistical analysis was performed using GraphPad Prism version 9.5.1 (GraphPad Software, San Diego, CA, USA).

## 3. Results

### 3.1. Study Population and Baseline Characteristics

We prospectively enrolled 64 AAV patients and 24 age- and vaccine-matched healthy controls before SARS-CoV-2 vaccination. Homologous or heterologous standard two-dose vaccination with mRNA (BNT162b2, BioNTech or mRNA-1273, Moderna) or adenovirus-vectored (ChAdOx1, AstraZeneca) vaccines were performed in all subjects and humoral responses as well as reactogenicity and AAV disease activity were assessed. Median (IQR) age at enrollment was 68 (55–72) years for AAV patients and 61 (59–64) years for healthy controls. Baseline characteristics and detailed vaccination regimens are shown in Table 1.

After a median (IQR) of 103 (72–126) and 198 (102–214) days after the second vaccine dose, a booster dose of BNT162b2 was administered to 21 AAV patients and 24 age- and vaccine-matched healthy controls, respectively. Homologous mRNA vaccination was performed in 19 (90%) and 19 (79%) and heterologous ChAdOx1/mRNA vaccination in 2 (10%) and 5 (21%) of the AAV patients and healthy controls, respectively. The baseline characteristics of subjects who received booster vaccination are shown in Table 2.

### 3.2. Anti-S1 IgG, Surrogate-Neutralizing Antibodies, and Antibodies to Various SARS-CoV-2 Epitopes in AAV Patients and Healthy Controls

After the initial standard two-dose vaccination, the humoral vaccine response was determined by anti-S1 IgG antibodies, surrogate-neutralizing antibodies, and antibodies to the full-spike protein and the RBD. In the AAV cohort, only 13/47 (28%) and 42/64 (66%) patients had detectable anti-S1 IgG antibodies after the first and second vaccine dose compared to 13/16 (81%) and 24/24 (100%) in healthy controls, respectively (Figure 2A). Anti-S1 IgG levels were significantly lower in AAV patients after the first and second vaccine dose as compared to healthy controls (*p* < 0.001 for both). In addition, with a median (IQR) percent inhibition of 69 (30–97) versus 96 (92–97), neutralizing antibody activity was significantly lower in AAV patients compared with healthy controls after the second (*p* < 0.001) whereas differences were not significantly different after the first vaccine dose (Figure 2A). Antibodies of 17/47 (36%) and 49/64 (77%) as well as 10/16 (63%) and 24/24 (100%) exceeded the threshold for neutralization in the surrogate neutralization assay after first and second vaccine dose in AAV patients and healthy controls, respectively (Figure 2A). Individual anti-S1 IgG and surrogate-neutralizing antibody courses from first to second vaccine dose are given in Figure 2B for each subject. Both parameters increased significantly from first to second vaccine dose in AAV patients and healthy controls.

In addition, healthy controls showed higher reactivity against different target epitopes determined by Luminex assays as compared to AAV patients after standard two-dose vaccination (Figure 2C). After second vaccination, 46/64 (72%) and 41/64 (64%) of AAV patients and 24/24 (100%) and 24/24 (100%) of healthy controls had MFI values above the cutoff for detection for the full-spike and the RBD protein, respectively (Figure 2C). AAV patients had significantly lower MFI values against the full-spike protein (median (IQR) 18,080 (6402–23,069) vs. 23,409 (22,864–23,975); *p* < 0.001) and the RBD protein (median (IQR) 10,312 (816–20,383) vs. 19,960 (17,451–21,455); *p* < 0.001) compared with healthy controls (Figure 2C). Antibodies to the nucleocapsid remained negative in all individuals, confirming the exclusion of previously infected participants. The individual distribution of antibodies to various SARS-CoV-2 epitopes in each subject is shown in Appendix A.

After the second vaccination, 39/64 (61%) AAV patients were triple positive for anti-S1 IgG, surrogate-neutralizing antibodies, and anti-RBD antibodies and 15/64 (23%) showed no humoral immune response as compared to 24/24 (100%) triple positivity in healthy controls (Figure 3A). The humoral immune response after second vaccine dose was further stratified for different immunosuppressive maintenance therapies of AAV patients. Anti-S1 IgG, surrogate-neutralizing antibodies, and anti-RBD antibodies were significantly higher in AAV patients receiving immunosuppressive maintenance therapy with steroids or mycophenolic acid/azathioprine compared with the anti-CD20 monoclonal antibody rituximab (*p* < 0.001 for all; Figure 3B). Only 9/28 (32%), 13/28 (46%), and 8/28 (29%) of AAV patients receiving rituximab maintenance therapy exceeded the cutoff for positivity for anti-S1 IgG, surrogate-neutralizing antibodies, and anti-RBD antibodies, respectively (Figure 3B).

### 3.3. Live-Virus Neutralization against the Omicron Subtypes BA.1 and BA.5 before and after Booster Vaccination

In 21 AAV patients and 24 healthy controls, BNT162b2 booster vaccination was administered. Both anti-S1 IgG and surrogate-neutralizing antibodies significantly increased after booster vaccination in AAV patients and in healthy controls (Figure 4A). Again, AAV patients showed significantly lower anti-S1 IgG and surrogate-neutralizing antibody values with a median (IQR) of 5.6 (0.5–150) and 56 (4–94) as compared to a median (IQR) of 202 (97–330) and 97 (90–98) in healthy controls, respectively (*p* < 0.001 for both).

Live-virus neutralization against the omicron subtypes BA.1 and BA.5 was additionally performed in all subjects after third dose vaccination. The ID_50_, indicating the serum dilution that reduces SARS-CoV-2 infection of cells by 50%, significantly increased in healthy controls for both, the BA.1 and the BA.5 variant (*p* < 0.01; Figure 4B). Overall, 23/24 (96%) and 22/24 (92%) of healthy controls showed neutralizing antibody activity in the live-virus assay. In contrast, neither the ID_50_ for the BA.1 nor for the BA.5 variant increased significantly in AAV patients after booster vaccination (Figure 4B). Only 8/21 (38%) and 0/21 (0%) of AAV patients had detectable neutralizing antibodies against the BA.1 or the BA.5 variant. The ID_50_ for the BA.1 and the BA.5 variant was significantly higher in healthy controls compared with AAV patient before and after booster vaccination, respectively (*p* < 0.001 and *p* < 0.01 for BA.1 and *p* < 0.01 and *p* < 0.001 for BA.5; Figure 4B).

### 3.4. Longitudinal Monitoring of Reactogenicity and Disease Activity of ANCA-Associated Vasculitis after a First, Second, and Third SARS-CoV-2 Vaccine Dose

Local and systemic reactions were assessed using a twelve-item questionnaire after first, second, and third vaccination in both cohorts (Figure 5). Local and systemic reactions increased from the first to the third vaccine dose in both cohorts and were more frequent in healthy controls as compared to AAV patients. Vaccination was generally well tolerated in all subjects, and no other adverse events were noted.

In addition, we monitored AAV disease activity and kidney function before SARS-CoV-2 vaccination as a baseline and during follow-up three weeks after the first, the second, and the third vaccine dose, respectively (Figure 6). During follow-up, no AAV flare was detected for all patients. Both PR3-ANCA and MPO-ANCA did not increase significantly after each vaccine dose as compared with baseline levels before vaccination (Figure 6A,B). Kidney function, determined by serum creatinine, eGFR, and proteinuria, was also stable during follow-up (Figure 6C–E). In two AAV patients, proteinuria increased after the second vaccine dose. However, levels decreased to baseline before and after the third vaccine dose, and neither serum creatinine nor ANCA levels increased in these individuals. PR3 levels increased in one patient after an initial mRNA vaccine dose, but there were no signs of AAV disease activity and kidney function remained stable.

## 4. Discussion

AAV patients are particularly susceptible to COVID-19 breakthrough infections and severe disease courses, possibly due to their advanced age at initial diagnosis combined with an extensive immunosuppressive therapy [23]. In addition, safety concerns have been raised regarding the use of highly immunogenic mRNA vaccines in autoimmune diseases such as AAV vasculitis, and relapses have been reported in several case series [20,24,25,26,27]. However, because AAV vasculitis is an orphan disease and patients have been excluded from most vaccine trials, data on SARS-CoV-2 vaccination are still limited and prospective vaccination studies that have focused exclusively on AAV patients are lacking. In this prospective observational cohort study, we comprehensively characterized both the neutralizing antibody activity against different omicron subtypes and the vaccine reactogenicity and disease activity in AAV patients longitudinally. The aim of this study was to provide additional insight into the immunogenicity, reactogenicity, and safety profile of currently available SARS-CoV-2 vaccines in AAV patients.

We show that AAV patients have significantly lower anti-S1 IgG and surrogate-neutralizing antibodies after first, second, and third vaccine dose as compared to healthy controls. These observations are consistent with those of Carruthers et al., where AAV patients had lower seroconversion rates after the first (64%) and second (83%) SARS-CoV-2 vaccine dose compared to the general UK population, respectively [28]. However, seroconversion rates were even lower in our study, which may be due to the older age of the patients and the use of an anti-S1 IgG assay instead of a pan-antigen assay including nucleocapsid epitopes [28]. Other studies reported comparable results of impaired humoral immune response, including not only AAV patients, but large and diverse groups of individuals with autoimmune inflammatory rheumatic diseases [29,30]. Following mRNA booster vaccination, we and others have shown that neutralizing antibody activity is significantly increased in most AAV patients [31,32,33]. However, patients treated with the CD20 monoclonal antibody rituximab had severely limited humoral immune responses even after booster vaccinations and high breakthrough infection rates have been reported for this cohort [7,34,35,36]. Based on expert opinion, an interval of at least 6 months between rituximab treatment and vaccination is recommended [37,38]. However, there are data showing a severely diminished humoral vaccination response even 9–12 months after the last rituximab infusion [39]. Based on these observations and currently available data, booster vaccination, even with adapted vaccines, cannot be recommended in individuals with B-cell depletion therapy.

Emerging omicron subtypes such as BA.1, BA.5, or XBB.1 are characterized by high transmissibility and advanced immune escape compared to wild type. AAV patients with lower neutralizing antibody levels may become particularly susceptible to these variants and a complete arrest of transmission does not appear to be possible with currently available vaccines and vaccination regimens. However, even if transmission cannot be completely prevented, there are several studies that confirm a significant correlation between neutralizing SARS-CoV-2 antibody levels and the frequency and severity of breakthrough infections in the general population and in immunocompromised individuals [4,40,41,42]. Against the alpha (B.1.1.7) and delta (B.1.617.2) variants, we and others have previously reported lower neutralizing antibody activity after standard two-dose vaccination and after booster vaccination [31,43]. However, there are no data investigating neutralization against the omicron variant with its different subtypes. In this study, we performed live-virus assays and show a strongly reduced neutralizing antibody activity against the BA.1 and especially against the BA.5 omicron subtype. Recently, adapted bivalent SARS-CoV-2 vaccines with BA.4/BA.5 epitopes have been shown to overcome immune escape and significantly increased neutralizing antibody activity against currently prevalent variants [44,45]. In immunocompromised individuals such as AAV patients with low neutralizing antibodies and a higher risk of breakthrough infections or severe disease courses, booster vaccination with adapted bivalent mRNA vaccines seems to be advisable, although no data are available yet.

An association between natural SARS-CoV-2 infection or vaccination and (re)activation of autoimmune diseases has been demonstrated by several research groups [20,24]. Endothelial dysfunction after SARS-CoV-2 infection or vaccination appears to be caused by several processes that include direct effects of virus–endothelial interaction as well as indirect hyperactivation of the immune system and paracrine signaling by infected epithelial cells [46,47,48,49]. Individuals with AAV, which is caused by excessive activation of neutrophils that subsequently release inflammatory cytokines and reactive oxygen species, and form neutrophil extracellular traps, may be particularly at risk for these vascular complications caused by SARS-CoV-2 spike proteins [50,51]. However, data on the safety of SARS-CoV-2 vaccines in patients with rare rheumatic diseases such as AAV are limited and further research is needed to confirm this potential interplay between AAV pathogenesis and endothelial dysfunction caused by SARS-CoV-2 infection or vaccination.

In our study, reactogenicity was monitored using a twelve-item questionnaire after the first, second, and third vaccine dose, respectively. Systemic and local adverse events occurred infrequently in AAV patients after all vaccine doses and were observed significantly more often in healthy controls after each vaccine dose. These data are consistent with our own observations in other immunocompromised individuals, such as hemodialysis patients or kidney transplant recipients, in whom intrinsic or extrinsic immunosuppression appears to reduce vaccine reactogenicity [12,13]. Several case reports have reported AAV flares after both SARS-CoV-2 vaccination and infection, and safety concerns have been raised regarding the use of highly immunogenic mRNA vaccines [24,26,52]. However, in the current study, no AAV patient experienced a disease flare and kidney function remained stable during follow-up. In a multicenter case-control study of 107 individuals with systemic vasculitis, including 57 AAV patients, Simoncelli et al. recently reported only one disease flare after a first mRNA SARS-CoV-2 vaccine dose and no further episodes after the second or third vaccination [53]. It should be noted that neither the study by Simoncelli et al. nor our current study are powered or designed to definitively exclude safety concerns in AAV patients after mRNA vaccination. Our study was also not powered to detect relatively rare but serious adverse events such as myocarditis caused by mRNA vaccines.

A limitation of our study is the lack of data on cellular immunity data after SARS-CoV-2 vaccination. Especially in some patients under rituximab therapy, it could be shown that in the absence of humoral immune response at least antigen-specific activation of cellular immunity was detected [18,32,54]. Nevertheless, it has been clearly demonstrated that neutralizing antibodies are considered highly predictive of protection against symptomatic SARS-CoV-2 infection [4]. In addition, our study did not detect anti-IgA or anti-IgM antibodies to the nucleocapsid antigen. Therefore, individuals with asymptomatic COVID-19 disease may have tested negative for anti-IgG antibodies to the nucleocapsid antigen and therefore may not have been identified as positive. Another problem in measuring humoral immunity after SARS-CoV-2 mRNA vaccination is the persistence of SARS-CoV-2 spike mRNA and protein in the germinal centers of lymph nodes for up to two months [55]. The presence of different spike epitopes at different time points may influence the breadth of the humoral immune response over time. Therefore, the timing for measuring humoral and cellular immunity after SARS-CoV-2 mRNA vaccination remains critical and can vary significantly in the first weeks to months after each vaccine dose. A general limitation of studies investigating the immunogenicity and efficacy of SARS-CoV-2 vaccines is that there are still no universally validated and accepted antibody thresholds that correlate with protection against breakthrough infections or severe COVID-19 disease courses. Another limitation of our study is the predominance of male AAV patients after the third vaccine dose.

In summary, this prospective cohort study demonstrates significantly lower live-viruses neutralization against omicron subtypes BA.1 and BA.5 after vaccination with previous SARS-CoV-2 vaccines in immunocompromised AAV patients. Different vaccine doses were generally well tolerated and no AAV relapses occurred during follow-up. These data indicate that novel vaccination strategies, including adapted bivalent mRNA vaccines against epitopes of emerging variants, are needed to further protect high-risk individuals such as AAV patients. However, in the subgroup of patients treated with B-cell depletion therapy, additional booster vaccinations or adapted vaccine doses cannot be recommended based on the current data available.

## Figures and Tables

**Figure 1 viruses-15-01778-f001:**
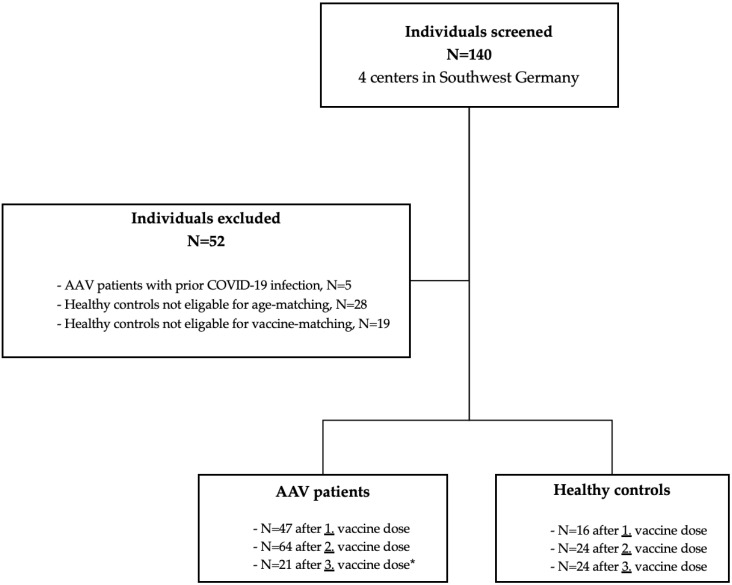
Flow chart of screened ANCA-associated vasculitis (AAV) patients and healthy controls after a first, second, and third vaccine dose. * 21 out of 64 AAV patients received a third vaccine dose at one of our Vasculitis Centers.

**Figure 2 viruses-15-01778-f002:**
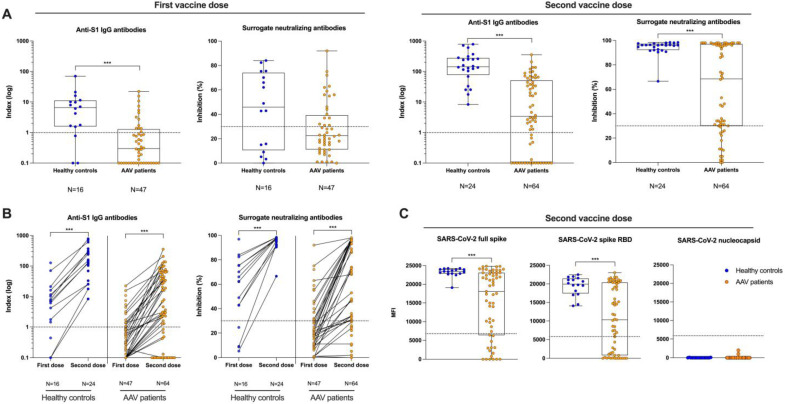
Anti-S1 IgG, surrogate-neutralizing antibodies, and antibodies to various SARS-CoV-2 epitopes after the first and second vaccine dose in ANCA-associated vasculitis patients and healthy controls. (**A**) SARS-CoV-2 S1 IgG antibodies plotted logarithmically as an anti-S1 IgG index in healthy controls and ANCA-associated vasculitis patients after first and second vaccine dose. The dashed black line represents the cutoff for detection. A semi-quantitative index of <1 was considered as negative. Neutralizing antibody capacity was measured in the same subjects after the first and second vaccine dose using a surrogate neutralization assay. The dashed black line represents the cutoff for viral neutralization in this assay according to the manufacturer’s instructions. A cut-off of <30% binding inhibition indicates absence of SARS-CoV-2 neutralizing antibodies and was defined as negative. (**B**) Paired anti-S1 IgG and neutralizing antibody courses after the first and second vaccine dose are shown (**C**). IgG antibodies against different SARS-CoV-2 epitopes, namely the full-spike protein, the receptor-binding domain (RBD) of the spike protein, and the nucleocapsid protein in healthy controls and ANCA-associated vasculitis patients after the second vaccine dose. The y-axis represents the mean fluorescence intensity (MFI) value of the reactivity. The dashed black line represents the cutoff for detection for each target, respectively. *** *p* < 0.001.

**Figure 3 viruses-15-01778-f003:**
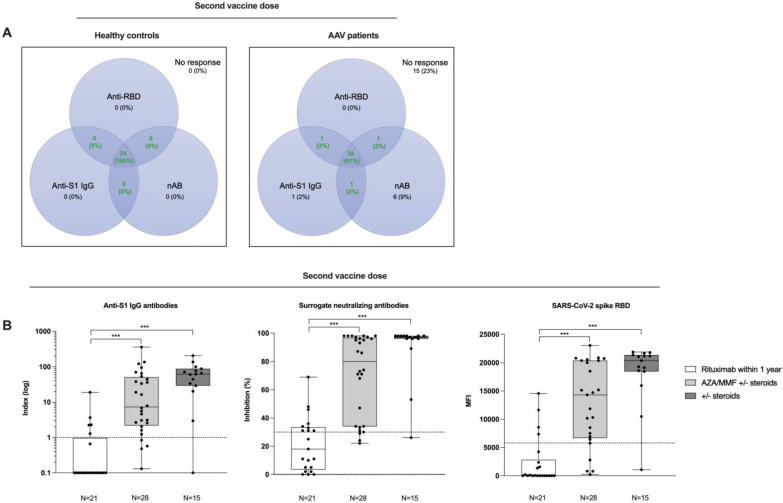
Seroconversion rates in healthy controls and in ANCA-associated vasculitis patients with different immunosuppressive maintenance regimens after second vaccine dose. (**A**) Seropositivity for anti-S1 IgG, surrogate-neutralizing, and anti-receptor-binding domain (RBD) antibodies after a second vaccine dose in healthy controls and in ANCA-associated vasculitis patients shown in a Venn-Diagram. Seropositivity was defined as an anti-S1 IgG index ≥1 in a chemiluminescent immunoassay, an inhibition ≥30% in a surrogate virus neutralization test, and a mean fluorescence intensity (MFI) ≥5800 in a bead-based multiplex assay. The green numbers in the middle of each panel indicate the proportion of subjects with seropositivity for two or three assays, respectively. (**B**) Anti-S1 IgG, surrogate-neutralizing, and anti- RBD antibodies in ANCA-associated vasculitis patients with different immunosuppressive regimens after a second vaccine dose. *** *p* < 0.001; AZA, azathioprine; MMF, mycophenolic acid; nAB, surrogate-neutralizing antibodies; RBD, receptor-binding domain.

**Figure 4 viruses-15-01778-f004:**
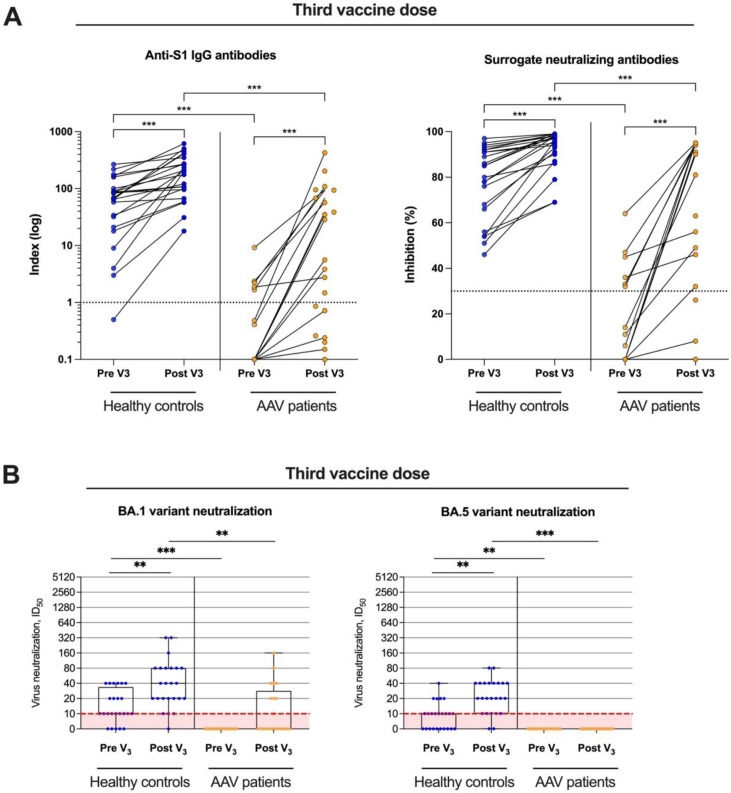
Live-virus neutralizing antibody activity against different SARS-CoV-2 variants in healthy controls and in ANCA-associated vasculitis patients after a third mRNA vaccine dose. (**A**) Paired anti-S1 IgG and surrogate-neutralizing antibody courses before (pre V3) and after (post V3) a third mRNA vaccine dose in healthy controls and ANCA-associated vasculitis patients. (**B**) Live-virus neutralization against the omicron subtypes BA.1 and BA.5 were performed using serum taken before (pre V3) and after (post V3) a third vaccine dose in 21 patients with ANCA-associated vasculitis patients and in 24 healthy controls. The ID_50_ on the y-axis indicates the serum dilution that reduces infection of cells by 50%. A neutralization titer of 1:10 is the cut-off for this test, indicated by a dashed red line. *** *p* < 0.001, ** *p* < 0.01.

**Figure 5 viruses-15-01778-f005:**
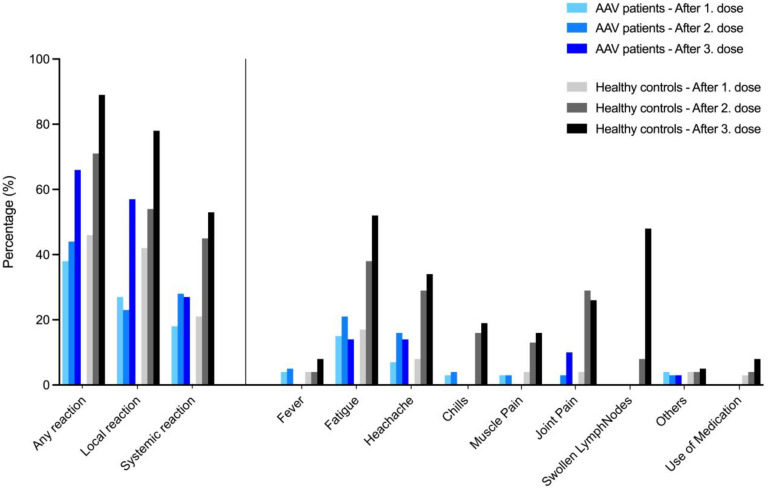
Local and systemic reactions after SARS-CoV-2 vaccination in healthy controls and in ANCA-associated vasculitis patients. Local and systemic reactions after a first, second, and third vaccine dose in healthy controls and in ANCA-associated vasculitis patients assessed by a questionnaire after each vaccine dose.

**Figure 6 viruses-15-01778-f006:**
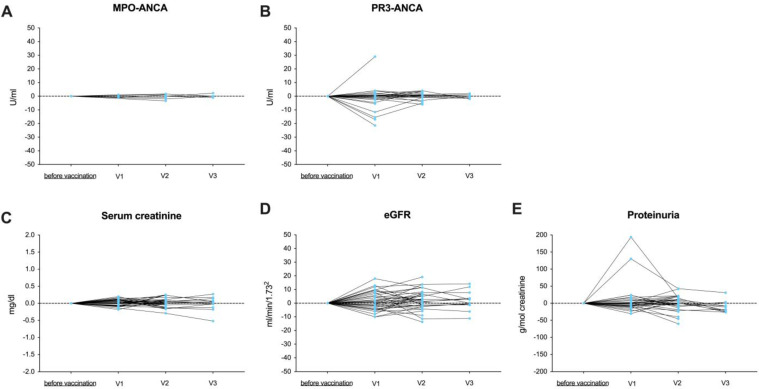
Disease activity detected by different ANCA antibodies and kidney function after SARS-CoV-2 vaccination in healthy controls and in ANCA-associated vasculitis patients. (**A**) MPO-ANCA and (**B**) PR3-ANCA were measured after first (V1), second (V2), and third (V3) vaccine dose. The antibody titer before SARS-CoV-2 vaccination was set as baseline (0), and differences from baseline were measured after each vaccine dose, respectively. Serum creatinine (**C**), eGFR (**D**), and proteinuria (**E**) were assessed after first (V1), second (V2), and third (V3) vaccine dose. Values before SARS-CoV-2 vaccination were set as baseline (0), and differences from baseline were measured after each vaccine dose, respectively. eGFR—estimated glomerular filtration rate; MPO—myeloperoxidase; PR—proteinase 3.

**Table 1 viruses-15-01778-t001:** Baseline characteristics of ANCA-associated vasculitis patients and healthy controls after first and second vaccine dose.

	AAV Patients (N = 64)	Healthy Controls (N = 24)	*p*
Age, median (IQR), y	68 (55–72)	61 (59–64)	0.58
Female sex, N (%)	34 (53)	14 (58)	0.81
Diagnosis, N (%)			
Granulomatosis with polyangiitis	47 (73)	-	-
Microscopic polyangiitis	15 (23)	-	-
Eosinophile granulomatosis with polyangiitis	2 (3)	-	-
Time since initial diagnosis, median (IQR), y	8 (5–14)	-	-
ANCA ELISA, N (%)			
PR3	44 (69)	-	-
MPO	12 (19)	-	-
Double positive	8 (13)	-	-
Organ involvement, N (%) *			
Ears, nose, throat	26 (43)	-	-
Kidney	49 (82)	-	-
Lung	45 (75)	-	-
Others	25 (39)	-	-
Induction therapy, N (%)			
CYC ± steroids	38 (59)	-	-
Rituximab ± steroids	26 (41)	-	-
Maintenance therapy, N (%)			
±steroids	14 (22)	-	-
Azathioprine or MMF ± steroids	22 (34)	-	-
Rituximab	28 (44)	-	-
Time since last Rituximab dose,	4 (2–7)	-	-
median (IQR), y			
Vaccination regimen, N (%)			
Homologous BNT162b2	49 (77)	19 (79)	0.99
Homologous mRNA-1273	4 (6)	0 (0)	0.57
Heterologous ChAdOx1/BNT162b2	6 (9)	2 (8)	0.99
Heterologous ChAdOx1/mRNA-1273	2 (3)	0 (0)	0.99
Homologous ChAdOx1	3 (5)	3 (13)	0.34

* Data not available for 4 patients. CYC—cyclophosphamide; IQR—interquartile range; y—years; MMF—mycophenolic acid.

**Table 2 viruses-15-01778-t002:** Baseline characteristics of ANCA-associated vasculitis patients and healthy controls after a third vaccine dose.

	AAV Patients (N = 21)	Healthy Controls (N = 24)	*p*
Age, median (IQR), y	71 (59–74)	61 (59–64)	0.16
Female sex, N (%)	7 (33)	14 (58)	0.14
Diagnosis, N (%)			
Granulomatosis with polyangiitis	13 (62)	-	-
Microscopic polyangiitis	8 (38)	-	-
Time since initial diagnosis, median (IQR), y	4 (1-10)	-	-
ANCA ELISA, N (%)			
PR3	11 (52)	-	-
MPO	8 (38)	-	-
Double positive	2 (10)	-	-
Organ involvement, N (%)			
Ears, nose, throat	4 (19)	-	-
Kidney	18 (86)	-	-
Lung	10 (48)	-	-
Others	4 (19)	-	-
Induction therapy, N (%)			
CYC ± steroids	18 (86)	-	-
Rituximab ± steroids	3 (14)	-	-
Maintenance therapy, N (%)			
±steroids	4 (19)	-	-
Azathioprine or MMF ± steroids	9 (43)	-	-
Rituximab	8 (38)	-	-
Time since last Rituximab dose,	3 (1–8)	-	-
median (IQR), y			
Homologous mRNA vaccination, N (%)	19 (90)	19 (79)	0.42
Heterologous ChAdOx1/2x mRNA, N (%)	2 (10)	5 (21)	0.42

CYC—cyclophosphamide; IQR—interquartile range; y—years; MMF—mycophenolic acid.

## Data Availability

The data of this study are available on request from the corresponding authors.

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
