# Peer review of "BA.1/BA.5 Immunogenicity, Reactogenicity, and Disease Activity after COVID-19 Vaccination in Patients with ANCA-Associated Vasculitis: A Prospective Observational Cohort Study"

_viruses, 2023, doi:10.3390/v15081778_

Round 1

Reviewer 1 Report

1.  These authors have prospectively studied the immune responses in patients with ANCA associated vasculitis following COVID-19 vaccination.  The assays include anti-S1 IgG, surrogate neutralizing antibodies, and live virus neutralization assays.  They also monitored these patients for adverse effects and reactivation of their underlying vasculitis.  This study included 64 patients and 24 healthy controls.  Assays were run after the first vaccine dose, the second vaccine dose, and the third vaccine dose.  None of the patients had a flare in their vasculitic disease.

2.  Patients with ANCA associated vasculitis had decreased immune responses to the first and second dose of vaccine.  In particular, they had significantly lower live virus neutralization against omicron subtypes after vaccination.  There were no significant changes in ANCA levels.  There were no significant changes in serum creatinine.  Two patients had an increase in proteinuria following the second vaccine which returned to their baseline after the third vaccine.
3.  This study provides important information about COVID-19 vaccination and chronically ill patients who have an underlying vasculitis and are often on immunosuppressive therapy.  The results indicate that these patients have a reduced immune response to vaccination but do not appear to have an increased number of adverse side effects or any flare in their underlying disease.  Studies like this are essential to understanding the effects of vaccination and to determine the adequacy of vaccination based on immune responses.

The English is fine

Author Response

Dear Reviewer,

thank you for the review and your positive feedback. The manuscript will be revised again for the English language.

Yours sincerely,

Claudius Speer

Reviewer 2 Report

The study by Speer et al is interesting, and could be useful for other clinicians.

However, I have several concerns that need to be addressed, looking at the data presented.

Not clear if the patients had infection before the doses (since this is the first time in history of medicine, or so, that we are vaccinating people that had a disease before).

The number of patients do not allow to detect adverse events like pericarditis/miocarditis, for instance. 

A new study from Berna’s University stated that 1 in 35 people had myocarditis after the COVID-19 vaccines the majority of which were women. My impression from that table is that females are under-represented in this study.

Many papers demonstrated toxicity of spike proteins for vascular cells and disruption of mitochondria, how this can be discussed for a disease which involved exactly inflammation of blood vessels?

What about troponin I and/or T? After the doses. The surveillance is not really active in these patients. Other papers showed rising levels of troponin in people after mRNA vaccines, do you have this parmater?

Why some patients were lost between the second and third dose? They did not make the 3 dose ? And if it is like that, why? 

What is exactly the follow up after the third dose? Regarding protection from disease, antibody increase and adverse events of any kind?

The response is very focused on spike reactivity. So this means that these patients never had covid19? Is it possible to change the scale on Figure 2 c for anti-N responses to see if a small response is present?

What about IgA?

The develppement of a vaccines for new variants is always after a while, this means that the protection will be always late.

Is it really sure that the only strategy is continuing with giving booster doses, which may have a cumulative effect? 

The authors should comment on the fact that spike mRNA and protein last in the lymphonodes several weeks, according to a Stanford study that is famous. In other study spike protein is circulating for at least three months. 

This may mean that more than one epitope (s) derived from spike are present at the same time, which can spoil the response.

Maybe the authors can discuss whether it is feasible to continue with booster at the time of Omicron, which increase the well-known adverse events after vaccination, while the protection is low and short leaving and, in particular, does not stop transmission. Stopping infection is highly desirable for these patients and other at risk patients, but it is clear that this goal cannot be achieved by these kind of vaccines judged by Anthony Fauci in a publication this year “largely unsuccessful”. Why than one should continue with boosters with these vaccines, when the seminal clinical trials involved two injections?

Figure 3 What was the point to vaccinate people taking rituximab which is a therapy that blocks B-cell? Indeed the response is scarce. Why than in the last part of the discussion the authors state that adapted vaccines will be suitable? Why adapted vaccines should be better when it is the use of rituximab that impairs the response? It is clear that these patients cannot benefit from other doses, even with adapted vaccines (which will be anyway always behind the new variants emerging).

Thus, the conclusions in the discussion contradict the data shown in the Figures.

Moreover, in the original Pfizer trial (I do not know for the others for sure, but I guess it is the same) no immunosuppressants should be taken by people to get the vaccines, since patients taking immune suppressive therapy were actually excluded from the trial, together with other subject having other issues.

Figure 4: there is statistics before and after vaccination in HD and AAV patients but it is clear that the vaccine effect was poor in AAV. One should compare the pre V3 and post V3 between HD and AAV. IF the difference is significant this should be commented.

Figure 6, the scale in figures could be changed to better see the effect of doses and AAV parameters. All the parameters where up-regulated in many individuals. Is it possible that in a subgroup of patients the dose were not well tolerated? There is any possibility to discuss why some patients had these parameters increased? Will these patients be excluded from a 4 dose (or second booster, as it is called)?

It is fine.
